# Does a SPR-Based Cell-Based Assay Provide Reliable Results on the Toxicity and Efficacy of Antiviral Drugs?

**DOI:** 10.3390/s25133905

**Published:** 2025-06-23

**Authors:** Petia Genova-Kalou, Evdokiya Hikova, Todor Kereziev, Petar Kolev, Vihar Mankov, Petar Veselinov, Trifon Valkov, Georgi Dyankov

**Affiliations:** 1National Center of Infectious and Parasitic Diseases, 44A “Gen. Stoletov” Blvd., 1233 Sofia, Bulgaria; petia.d.genova@abv.bg; 2Institute of Optical Materials and Technologies “Acad. J. Malinowski” (IOMT), Bulgarian Academy of Sciences (BAS), 109 “Acad. G. Bonchev” Str., 1113 Sofia, Bulgaria; p.kolev@iomt.bas.bg (P.K.); vmankov@iomt.bas.bg (V.M.); pveselinov@iomt.bas.bg (P.V.); gdyankov@iomt.bas.bg (G.D.); 3Central Laboratory of Applied Physics, Bulgarian Academy of Sciences, 61 Sankt Petersburg Blvd., 4000 Plovdiv, Bulgaria; todorkrz@gmail.com; 4Prof. Ivan Kirov Hospital, Department of Infectious Diseases, Medical University of Sofia, 1431 Sofia, Bulgaria; t.valkov@medfac.mu-sofia.bg

**Keywords:** surface plasmon resonance, cell-based assay, toxicity, antiviral activity, drugs

## Abstract

**Highlights:**

**What are the main findings?**
A SPR-based cell-based assay provides accurate data on drug cytotoxicity and antiviral efficacy.Grating-based SPR requires sequential signal measurements of SPR slides removed from the medium at fixed hours after seeding, which successfully replaces continuous-flow SPR detection.

**What is the implication of the main finding?**
Exploiting the key advantage of grating-based SPR—tuning the excitation wavelength of the resonance from the visible to the near-infrared region—to detect cell morphological changes with increased accuracy and sensitivity;Speed of detection: The SPR assay detects the precursors of biochemical reactions in cells before 48 h, which are detected by the MTT assay at 96 h.

**Abstract:**

SPR has been recently established as a powerful tool for studying various cellular processes in real time and without the use of labeling agents. So far, all studies in this area have been performed using the Kretschmann method for SPR excitation. In our studies, we used grating-based SPR. Here, we investigated the feasibility of this approach in a cell-based assay applied for antiviral drug screening. It was found that the continuous-flow SPR detection used in the conventional SPR can be replaced by sequential signal measurements of SPR slides removed from the medium at fixed hours after seeding. A protocol ensuring correct measurements was established. SPR detection was performed up to 48 h after seeding the VERO E6 cell line in three experiments, in which the cells were (i) compound-untreated, (ii) compound-treated, and (iii) infected with human coronavirus type 229E and compound-treated. Therefore, the temporal variation in the SPR signal was determined, induced by the cell coverage on the slide, the compound toxicity, and its antiviral action. MTT analysis and microscopic observations were used as reference methods. The remarkable agreement found in the results of SPR detection proved the effectiveness and reliability of grating-based SPR applied in cell-based assays.

## 1. Introduction

SPR is conventionally applied to detect biomolecular interactions and is of paramount importance for fundamental biological research. The need to study protein interactions at a cellular level has driven the development of new measurement techniques, including SPR imaging, nanoplasmonics and microfluidics, phase-based SPR detection, plasmon waveguide resonance spectroscopy, and signal-locked SPR. A comprehensive analysis of the advantages and disadvantages of these techniques can be found in [1]. SPR is also widely applied for health science research, drug discovery, clinical diagnosis, and environmental and agricultural monitoring. Achievements in these fields are summarized in [2]. Recent applications and perspectives of SPR technology are highlighted in [3].

Identifying biochemical interactions is not enough to determine their significance in life processes. Their physiological significance must be assessed in vivo or in vitro. Cell-based assays are a relevant screening system for this purpose. In recent years, SPR has been successfully applied in such assays.

The penetration depth of the plasmon wave increases from approximately 150 nanometers for SPR in the visible range to approximately 400 nanometers for SPR in the near-infrared range. This distance is from the metal surface on which the cells are cultured and represents only a small fraction of the height (vertical dimension) of commonly used cells, which is in the range of several micrometers. Therefore, the SPR signal is thought to be triggered by biological events in the region close to the cell membrane [4,5,6].

It has been shown [7,8,9] that changes near the plasma membrane may reflect the accumulation and rearrangement of proteins activated by intracellular signaling triggered by exogenous stimuli. Thus, complex biological events in cells cause local changes near the membrane and generate an SPR signal.

The same mechanism generates the signal in the Raman spectroscopy of cells. This powerful technique is used to study cellular biochemistry. Similarly to SPR, Raman spectroscopy can provide information about time-dependent changes in cellular biochemistry without the need for cell labeling or staining. It can be used to observe how cells interact with drugs and determine the statistical effects of drug exposure in real time, which is important for assessing the efficacy of the drug being tested. The same observations can be made using SPR-based cell research [10].

Unlike SPR, Raman spectroscopy can identify specific biochemical reactions within different structural elements of the cell, making it a powerful method for studying cellular biochemistry. This requires complex equipment and precise data processing based on an extensive database. The SPR-based cell assay has a cheaper and more accessible instrumental base, and its results accurately determine morphological changes and cell proliferation.

The demand for cell-based functional assays is increasing, as they have demonstrated significant potential for development. The main advantage of cell-based assays is that they offer more relevant screening systems from a physiological point of view. These assays are employed in the identification of optimal drug candidates and alterations in morphology, measurement of proliferation, analysis of cell signaling pathways, and assessment of toxicity.

The methyl thiazolyl tetrazolium (MTT) assay is the most commonly used method for testing viability of virus-infected and mock-infected cells and their proliferation after exposure to antiviral drugs. The assay is predicated on the capacity of mitochondrial dehydrogenase enzymes in viable cells to reduce yellow MTT upon cleavage of the tetrazolium ring to a water-insoluble purple–blue entity, which is determined spectrophotometrically [11].

Despite its limitations, including those related to seeding density, sensitivity to environmental conditions, and dependence on cellular metabolism [12], the MTT assay has been used extensively for antiviral screening of new drugs.

It should be noted that there are only a limited number of articles available which report on the use of SPR-based drug screening methods involving a cell-based assay [13,14,15]. These articles include a variety of drugs that have been tested.

In our previous work [16], we demonstrated that the SPR-based cell assay is a reliable tool for studying host cell–virus kinetics and antiviral drug activity. In the years that followed, the methodology developed was refined for SPR-based cell screening of antiviral drugs. It should be noted that, unlike all similar studies in the field, we use grating-based SPR. Since SPR is excited on the analyte side (i.e., the grating is illuminated through the cells), it is difficult to use a controlled flow of the analyte. This requires a modification of the adopted methodology for SPR-based cell analysis. In order to achieve this, sequential measurements of the SPR signal must be taken at fixed hours after cell seeding, corresponding to different stages of the cell life cycle. In order to proceed, it is necessary to study the SPR signal induced by morphology changes/cell proliferation. Although this problem has been discussed in [17,18] to some extent, the results are controversial, probably due to the cell growth specificity of a particular cell line.

The objective of the present study was to evaluate the reliability of SPR-based cell assays in the context of antiviral drug screening, with a particular focus on the characteristics of grating-based SPR. MTT assay was used to provide reference data.

## 2. Materials and Methods

### 2.1. SPR Slides

The SPR was excited in diffraction gratings with 1500 nm period and 70 nm groove height, with a multilayer metallic coating deposited on them, the last layer being gold. Prepared in this way, they served as SPR slides. The key advantage of diffraction grating is that it allows for straightforward tuning of the SPR wavelength by altering the incident angle, ranging from approximately 610 nm (at 80 deg) to around 1030 nm (at 41 deg). This allows the penetration depth of the plasmon wave field to be adjusted from about 300 nm (in the visible range) to about 520 nm (in the near infra-red range), and, consequently, for the detection sensitivity to be varied.

### 2.2. SPR Measurement and Setup

The SPR slides were mounted on a Θ-2θ goniometer, so that the incidence and reflection angles could be precisely controlled within the range of 41–80 degrees, with an accuracy of 0.01 degrees. A collimated beam of p-polarized white light was used to excite SPR resonances. Spectral interrogation was employed to monitor SPR in the zero order of reflection using an Avasoft AVASPEC-ULS2048CL-EVO spectrometer. Please refer to [16] for more detailed information on our SPR measurement system.

### 2.3. Cells and Viruses

For the evaluation of cytotoxicity and the antiviral experiments, VERO E6 cell line was utilized, derived from African green monkey kidney sub clone E6 and kindly provided from the Cell Collection of NCIPD, Sofia (Bulgaria). The cells were cultured in Dulbecco’s Modified Eagle’s Medium (DMEM, Sigma-Aldrich, Sent Luis, MO, USA), supplemented with 10% heat-inactivated fetal bovine serum (FBS) (Gibco, Life Technology, Darmstadt, Germany) and antibiotics (penicillin, streptomycin). The cultures were maintained at 37 °C in a humid atmosphere at 5% CO_2_ (standard conditions) and trypsinization was used for cell maintenance. The cells were seeded at a density of 5 × 10^4^ cells/mL, and cell monolayers were used for the experiments after a growth period of 24 h.

Human coronavirus type 229E (HCoV-229E) stock (5 log TCID50/mL) was propagated in Vero E6 cells in DMEM media supplemented with 2% FBS at 33 °C, 5% CO_2_. Viral titers were determined by endpoint dilution assays on Vero E6 cells and stored at—80 °C until use.

### 2.4. Drug Preparations

Chalcone derivatives were diluted in dimethyl sulfoxide (DMSO, Sigma-Aldrich, 82024 Taufkirchen, Germany) at a concentration of 10 mg/mL, based on their molecular weight. The 1 mg/mL stock was further diluted in cell culture media at the time of use, at concentrations ranging from 0.0001 to 10 mg/mL.

### 2.5. SPR-Based Cell Assay

Before seeding the cells, the slides were immersed in isopropyl alcohol and cleaned ultrasonically for 10 min. Then they were rinsed thoroughly with high purity water, dried, and sterilized by UV light for 24 h.

The slides were inoculated in polycarbonate 12-well cell culture plate and Vero E6 cells were seeded and cultured at 37 °C, 5% CO_2_ atmosphere at densities 4 × 10^4^, 2 × 10^4^, and 1 × 10^4^ cells/mL and incubated for 6 h, 9 h, 24 h, 30 h, 33 h, and 48 h to allow cell adhesion to the SPR slide surface.

After a certain period of time, the slides seeded with different cell concentrations were extracted from the medium, gently rinsed with deionised water, and subjected to centrifugation to facilitate the separation of the liquid phase. Subsequent to this, a spectral measurement of the SPR was conducted, and the cell number and confluency were determined through microscopic examination.

### 2.6. MTT Assay for Cytotoxicity Evaluation

The cytotoxicity of the compounds was determined by the microscopic observation of monolayers and by MTT. Vero cells (5 × 10^5^ cells/mL) were treated with different concentrations of the compound (0.0001–10 mg/mL). The optical density (OD) was determined at *λ* = 540 nm by an ELISA reader (Bio-Tek Instruments GmbH, 74177 Bad Friedrichshall, KOCHENDORF, Baden-Württemberg Germany). The percentage (%) of viable treated cells was calculated in relation to untreated controls (OD_exp_/OD_cell control_ × 100). CC_50_ was the concentration required to reduce the OD or to induce visible morphological changes in 50% of the cells. The maximum nontoxic concentration (MNC) was defined as the highest concentration which did not cause injury or death to the treated cells.

### 2.7. MTT-Based Antiviral Assays

The antiviral activity of the compounds studied was determined by the MTT test proposed in [19].

For our experiment, we used a viral suspension of 100 TCID50 (50% infectious dose for tissue culture), following the method described in our previous work [20].

In the experimental setup, the plates were processed as follows:Control cells (not infected with virus and not treated with compounds)—0.2 mL of supporting cell culture medium (2% FBS) were added to the wells designated for cell control;Virus control (virus-infected and compound-untreated cells)—0.1 mL of supporting cell culture medium with the appropriate virus concentration were added to the wells designated for virus control;Cells exposed to the compounds—infected with a virus and treated with different concentrations of the compounds studied—0.1 mL of the dilutions of the substances prepared in advance.

### 2.8. Cell Coverage Experiment

The changes in the cell coverage on the SPR slide can be attributed to two main factors: an increase in cell number due to cell growth or morphological changes in the cell layer. Cell coverage is the sole factor determining the spectral shift in the SPR when cells are not subjected to any external influence. The shift, measured relative to a bare SPR slide and termed “cell control”, was established by following the procedure outlined in Section 2.5.

### 2.9. Microscope, Cell Number and Confluence

In order to ascertain the relationship between cell coverage of the slide and SPR response, the number of cells and the confluence of the cell layer on each SPR slide were determined. To this end, an Axiovert 5 inverted microscope and Image software (Zeiss Labscope 4.5) with AI capabilities were utilized. Each slide was placed in a well with a medium and the number of cells on it were microscopically counted; counting was performed once again after the SPR measurement. For each slide, measurements were performed in an area that had been previously marked. Focusing the image just above the surface of the slide and on the cells was imperative, in order to avoid counting artifacts misidentified by the AI as cells.

## 3. Results

### 3.1. Cell Control: SPR Signal, Confluence, and Cell Coverage

As [17] has explained, increasing cell coverage leads to an increase in resonance shift. Ref. [21] has provided an interpretation of the change in the SPR signal in drug–cell interaction after monolayer formation, demonstrating that signal changes are solely due to the interaction under study. This approach is feasible in the conventional case of the SPR-based cell assay with continuous flow, where the changes in the cells are observed in real time. In our case of grating-based SPR, we take snapshots of the cell growth, as it is necessary to distinguish the morphology/confluence-induced SPR signal from the signal induced by the drug–cell interaction. For this purpose, as presented in Section 2.8, we define a reference signal called “cell control”. Figure 1 shows cell control at different cell seeding densities depending on cell seeding time.

Figure 2 shows the number of cells on the SPR slide as determined by observation under microscope using AI software (Zeiss Labscope 4.5). The comparison with Figure 1 shows a good correlation with the SPR signal, which can be considered to be induced by the cell coverage. Note that this is the SPR signal in the NIR region.

At a seeding concentration 4 × 10^4^ cells/mL, the resonance shifts increased with cell seeding time. At concentrations 2 × 10^4^ cells/mL and 1 × 10^4^ cells/mL, the resonance shifts demonstrated minimal change over time. At a seeding concentration of 4 × 10^4^ cells/mL, the number of cells increased continuously with cell seeding time as a result of active proliferation. Microscopic observation demonstrated an increase in the confluence of the layer, from 30% at 24 h to 82% at 48 h. At concentrations of 2 × 10^4^ cells/mL and 1 × 10^4^ cells/mL, the number of cells detected by microscopic observation remained consistent over time. This indicates low proliferation, as evidenced by the low confluence observed microscopically.

Figure 3 shows the SPR signal at 650 nm measured on the same slides, on which the SPR signal was measured in the infrared range. Comparison with Figure 1 shows the low sensitivity of measurement.

An increase in cell coverage at a seeding density of 4 × 10^4^ cells/mL (Figure 2) causes a change in the SPR signal, which is indicative of the sensitivity of detection. However, the same change in cell coverage causes different changes in the SPR signal in different spectral ranges. In the visible range (Figure 3), the change is less than 30 nm over 30 h of seeding time and indistinguishable from the SPR signal induced by a seeding density of 2 × 10^4^ cells/mL. Furthermore, time dependence does not follow that of cell coverage. In the near-infrared (NIR) range (Figure 1), the change in the SPR signal is greater than 40 nm and gradually increases for up to 30 h. After this time, the signal saturates upon the formation of a cell monolayer.

This analysis illustrates the lower detection sensitivity within the visible range. This is probably due to the low penetration depth of the evanescent field. When the monolayer of cells is not formed yet, the evanescent field senses artifacts trapped between the grooves of the diffraction grating. These artifacts were very visible under the microscope, and to avoid them when counting cell number and assessing confluence, focusing was performed above the slide surface. In SPR detection in the visible range, their relative proportion in the signal was greater than that of the cells due to the distribution of the penetration field.

The limit of detection (LOD) of our measurement system was determined by conducting experiments with solutions of extremely low concentrations. It has been established [22] that the LOD is approximately 120 fM, corresponding to a spectral shift of around 1.8 nm that can be reliably measured. As the observed cellular response is in the order of tens of nanometers, LOD does not affect the sensitivity of the measurement.

Another factor affecting the time dependence of the SPR offset was the washout of the slides after removal from the medium. Even if performed carefully, this changed the number of adherent cells. Microscopic observation revealed a 10% reduction in cell number after washing at a concentration of 4 × 10^4^ cell/mL, whereas at concentrations of 2 × 10^4^ cell/mL and 1 × 10^4^ cell/mL, the reduction in cell number was approximately 50%. This phenomenon can be explained by the aforementioned lack of confluence at low cell seeding. As shown in Figure 2, there was a significant difference in the number of adherent cells at density 4 × 10^4^ cells/mL and 1 × 10^4^ cells/mL 48 h after seeding. This was confirmed by the microscopic observations as shown in Figure 4.

It is important to note that the 10% reduction in the number of adherent cells (at seeding density 4 × 10^4^ cells/mL) after washing does not significantly alter the temporal variation in cell number. It is consistent with the temporal variation observed in the medium. This finding suggests that the minimum seeding density should be set at 4 × 10^4^ cell/mL to ensure the formation of a confluent layer, favorable for antiviral drug screening and providing a reliable SPR signal.

Another advantage of the diffraction grating is that cells adhere more easily to a scalloped surface than to a smooth surface. This was demonstrated by the microscopic observation of the cell coverage on a diffraction grating and on gold-coated glass substrates at the above-mentioned hours after cell seeding. To ensure uniform properties of the metal coatings on the SPR slide and the glass substrate, they were deposited simultaneously in one run. Washing was found to reduce the number of cells on the smooth surface by more than 80%.

### 3.2. SPR-Based Evaluation of Cytotoxicity

The next step was to evaluate the potential of SPR-based drug screening, which was the primary objective of the present study. For that purpose, the cytotoxic and antiviral properties of synthetic chalcone derivatives with various substitutions in their aromatic nuclei were investigated on HCoV-229E in cell cultures. The cellular response to chalcones was measured at the cellular level in order to determine the level of sensitivity of the cells.

The structural formula of chalcone compounds is 1,3-diaryl-2-propen-1-one, alternatively designated as chalconoid, and they constitute a substantial group of flavonoids, exhibiting a wide spectrum of pharmacological action. Notably, these compounds are non-toxic and non-mutagenic to healthy cells, a property that renders them an attractive subject of research. Following extensive clinical investigations, numerous chalcones have been formally approved as effective pharmaceutical agents, as well as food supplements and ingredients of cosmetic products.

The cytotoxic and antiproliferative activities of AV6 and AV12 chalcone derivatives were determined on Vero E6 line cells cultured on SPR slides. The cell culture was treated at the time of seeding with chalcones of varying concentrations. At the 6th, 9th, 24th, 33rd, and 48th h after seeding, the slides were removed from the cell culture medium and measured as described in Section 2.5. The SPR signal was then read against the cell control. The results obtained are presented in Figure 5.

The deviation from the cell control can be attributed to the morphological changes induced by the chalcones in the cells. A reduction in deviation is indicative of reduced toxicity. The root mean square deviation (RMSD) for the various chalcone concentrations was as follows: AV6 (2.5 mg/mL)—6.417; AV6 (1 mg/mL)—3.346; AV12 (0.007 mg/mL)—1.098. It is evident that AV12 exhibited the lowest level of toxicity. The pronounced cytopathic effect was indicative of substantial RMSD value, as shown in Figure 4. As outlined in the MTT study, for AV6, the concentrations 2.5 mg/mL and 1 mg/mL were the cytotoxic concentration CC_50_ and the maximal non-toxic concentration, respectively; for AV12 the concentration of 0.007 mg/mL was the maximal non-toxic concentration.

### 3.3. MTT-Based Evaluation of Cytotoxicity

MTT is a quantitative, rapid, and highly reproducible method that has been accepted as the gold standard for assessing the cytotoxic effect of a wide range of therapeutic substances and drugs. The survival rate was reported at 96 h following treatment, as this time interval is characterized by significant cell proliferation. The MTT assay of cell survival and proliferation was conducted over a wide range of concentrations of the chalcones studied, from 0.0001 to 10 mg/mL, in order to accurately establish the MNC values. The untreated Vero E6 cells, which were assumed to have 100% survival rate, served as negative control. The MTT assay was used to obtain dose-dependent curves of the effect of chalcone derivatives on cell monolayer viability. The observed dependencies for AV6 and AV12 are shown in Figure 6.

The concentrations of 1 mg/mL and 0.007 mg/mL for AV6 and AV12, respectively (Figure 6), correspond to the MNC, as shown in Figure 6. The GraphPad Prism software version 8.00 for Windows, La Jolla California USA (www.graphpad.com (accessed on 17 June 2025)) was applied for preparation of the graphs and the statistical analysis of the data based on Two-Way ANOVA. Experiments were performed in triplicate, an independent sample Student’s *t*-test was used, and *p* < 0.05 was considered significant. Maximum nontoxic concentration (MNC) and cytotoxic concentration 50% (CC_50_) were obtained from dose–effect curves. The correlation coefficient values (R^2^) were calculated.

The results of the SPR-based analysis conducted up to the 48th h demonstrated a high degree of consistency. It is postulated that the fluctuations in the SPR signal are attributable to alterations in the membrane or the cytoplasm in close proximity, given that proliferative processes are incorporated within the cell control framework. This observation underscores the remarkable sensitivity of SPR-based assays. RMSD can provide a quantitative estimate of the degree of the cytopathic effect, but only after a comparative evaluation with the results of MTT in a large number of drug preparations. Nevertheless, a relative assessment of the cytopathic effect of one drug compared to another can be performed with ease and expediency.

### 3.4. SPR-Based Evaluation of Inhibitory Effect

In this experiment, a number of SPR slides with the cell culture were infected with HCoV-229E virus 24 h after seeding Vero E6 cells (5 × 10^4^ cells/mL). Half of these slides served as a viral control (VC). To assess the impact of the chalcone derivatives under study on the replication of HCoV-229E, the remaining half of the slides were treated with AV6 and AV12 at the MNC simultaneously with infecting the cells. At predetermined intervals post infection (p.i.) treatment, the slides were removed from the cell culture medium and subjected to the procedures outlined in Section 2.2 and Section 2.5.

Figure 7 shows the inhibitory effect of the AV6 and AV12 treatment on the virus replication cycle. The deviation from the viral control was particularly evident for AV6, especially after 26 h (p.i.). This finding suggests that the treatment effectively suppressed viral development after this time point. In contrast, the inhibitory effect of AV12 was not as evident throughout the entire observation period. Treatment with this chalcon derivative induced an SPR signal that differed little from that of the viral control. The observed low toxicity of AV12 is likely to be associated with its minimum biochemical activity, which is also evident in its low inhibitory potential.

Unlike SPR caused by cell coverage, the SPR response induced by the antiviral action of drugs was much weaker—the spectral shift did not exceed 10 nm. Therefore, a spectral shift of 1.8 nm, which corresponds to the established limit of detection (LOD) of our measurement system, affects the detection of biological events occurring near the cell membrane. It should be noted that LOD assessment in biomolecular interactions may differ from that in an SPR-based cell assay. The LOD depends on the analyte and is likely to differ for different cell lines. To our knowledge, no studies have examined the LOD in SPR-based cell assays. However, numerous studies have conclusively demonstrated that detection sensitivity increases in the IR range in both SPR applications: molecular recognition and cell-based assays.

The root mean square deviation (RMSD) can be employed as a quantitative metric to characterize the inhibitory effect, with larger values indicating greater effectiveness of the drug preparation. The utilization of RMSD facilitates a straightforward comparison of the efficacy of one pharmaceutical agent relative to another, as evidenced in the present case. However, it is challenging to define RMSD values that accurately quantify drug effectiveness. This objective could be achieved through experimental studies of numerous drug preparations, which would provide reliable statistical information.

### 3.5. MTT-Based Evaluation of Antviral Activity

To determine the anti-HCoV-229E activity of the chalcone derivatives tested, concentrations around and close to the minimum effective concentration (0.0001–1 mg/mL) were used. The MTT assay determined the level of protection after treatment with the chalcones relative to the viral control. These results are shown in Figure 8.

Based on these data, we concluded that the low toxicity of the AV6 chalcone derivative, when administered alone, inhibited HCoV-229E replication in a dose-dependent manner and effectively inhibited viral infectivity in the concentration range of 0.001–1 mg/mL. When, administered in MNC (1 mg/mL), it reduced viral yield (90% cellular protection) and the AV12 chalcone derivative did not affect the viral infectivity yield (35% cell protection).

The difference observed in the antiviral efficacy of the chalcones was fully consistent with the inhibitory potential of AV6 and AV12 determined by SPR.

## 4. Discussion

The cell coverage experiment showed that the SPR signal is determined only by the number of cells covering the SPR slide. The number of cells is in turn determined by (i) their ability to adhere, which is better on a scalloped surface, such as the diffraction grating used in our experiment; (ii) confluence, which is pronounced at higher cell seeding densities. Slide washing, which is mandatory for SPR measurements, always reduces the number of adherent cells, but this has little effect on a confluent layer formed at cell seeding densities greater than 4 × 10^4^ cells/mL. It is therefore advisable to work at densities that ensure reliable SPR measurements.

Cytotoxicity studies using SPR and MTT analyses showed identical results. The MTT assay detected cytotoxicity at 96 h post treatment. Apparently, the SPR assay effectively detected the precursors of the biochemical reactions occurring at a point of time later than 48 h. This can be interpreted as an advantage of SPR analysis. With the SPR assay, it is suggested that cytotoxicity should be characterized by the root mean square deviation relative to the cell control. RMSD can provide a quantitative assessment of the relative cytopathic effect of one drug compared to another, which is easily and rapidly achieved by SPR analysis.

The positive correlation between the results of SPR and MTT assays regarding the inhibitory effect of the chalcone derivatives also validates the efficacy of SPR detection of the precursors of biochemical reactions. The outcome of these reactions dictates the percentage of cell survival as determined by MTT at 96 h. The temporal changes observed in the SPR signal remain unexplained so far, since the biochemical mechanism of AV6 affecting viral replication during the initial hours of infection have not been elucidated yet.

## 5. Conclusions

In the present study, we established the characteristics of an SPR-based cell-based assay, in which the plasmon wave was excited in a diffraction grating. We exploited the main advantage of grating-based SPR—tuning the excitation wavelength of the plasmon from the visible to the near-infrared region, in order to detect cell morphological changes with increased accuracy and sensitivity.

It was shown that continuous flow measurements, applicable in conventional SPR-based cell analysis, were successfully replaced in grating-based SPR by sequential measurements at fixed hours after cell seeding. The latter were performed after removal of the SPR slides from the medium, as this procedure had been established as ensuring reliable measurements.

The consistent results obtained in the comparative drug screening studies conducted using SPR and MTT analyses demonstrate the effectiveness of SPR-based cell-based assay, where the plasmon wave is excited in diffraction gratings.

## Figures and Tables

**Figure 1 sensors-25-03905-f001:**
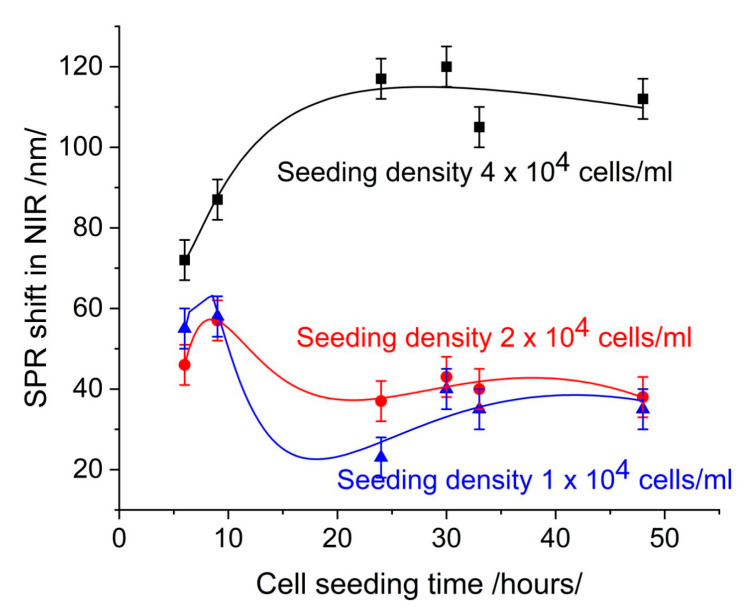
The morphology/confluence-induced SPR shift: cell control.

**Figure 2 sensors-25-03905-f002:**
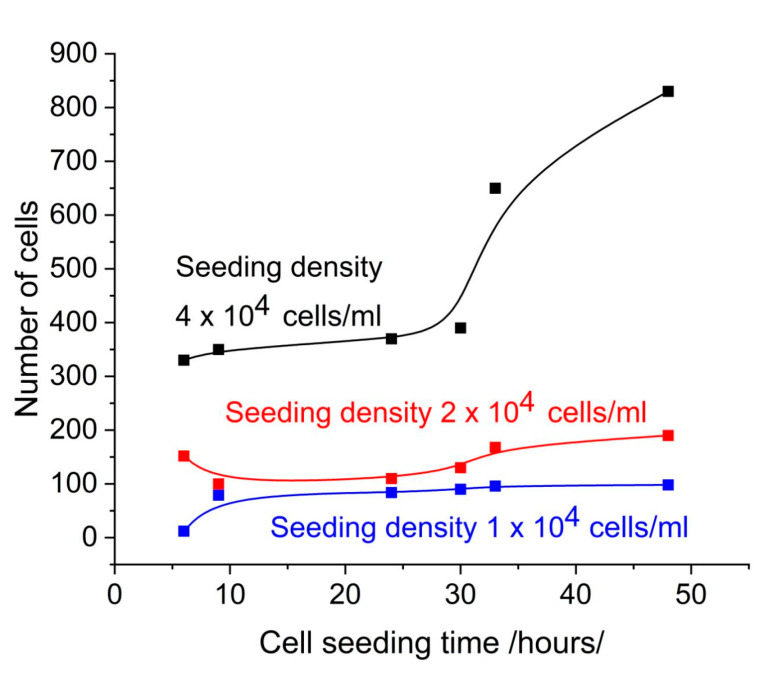
Number of cells on the SPR slide as detected by microscopic observation.

**Figure 3 sensors-25-03905-f003:**
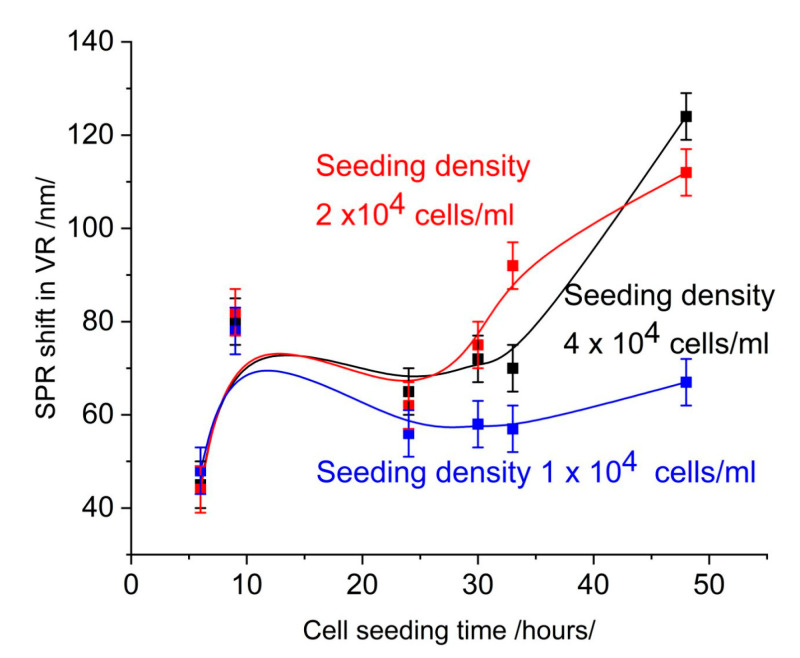
SPR signal in the range of 650 nm.

**Figure 4 sensors-25-03905-f004:**
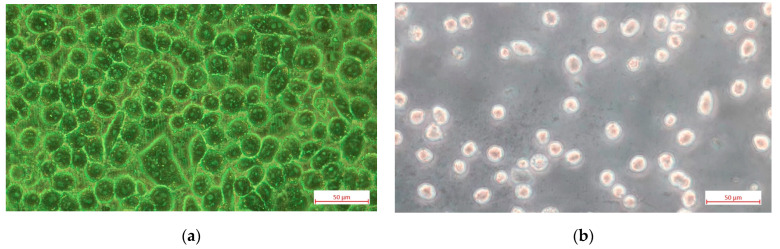
Light microscopy images of VERO E6 cells cultured on an SPR sensor slide with a cell seeding density: (**a**) 4 × 10^4^ cells/mL; (**b**) 1 × 10^4^ cells/mL at 48 h after cell seeding.

**Figure 5 sensors-25-03905-f005:**
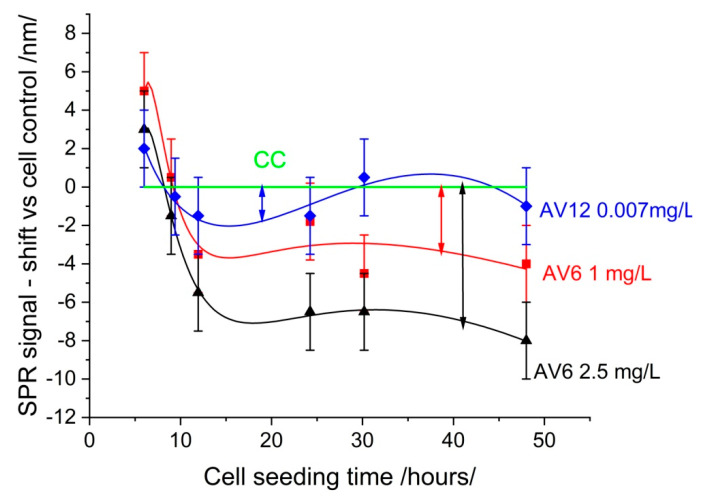
SPR signal variations in cellular response to chalcones AV6 and AV12 at a variety of concentrations. The green signal was cell control.

**Figure 6 sensors-25-03905-f006:**
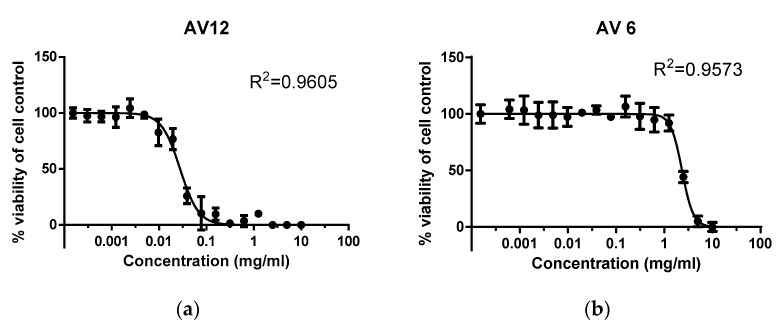
Effect of drugs on survival/proliferative activity of Vero E6 cell line as determined by MTT assay after 96 h of treatment with: (**a**) AV6; (**b**) AV12.

**Figure 7 sensors-25-03905-f007:**
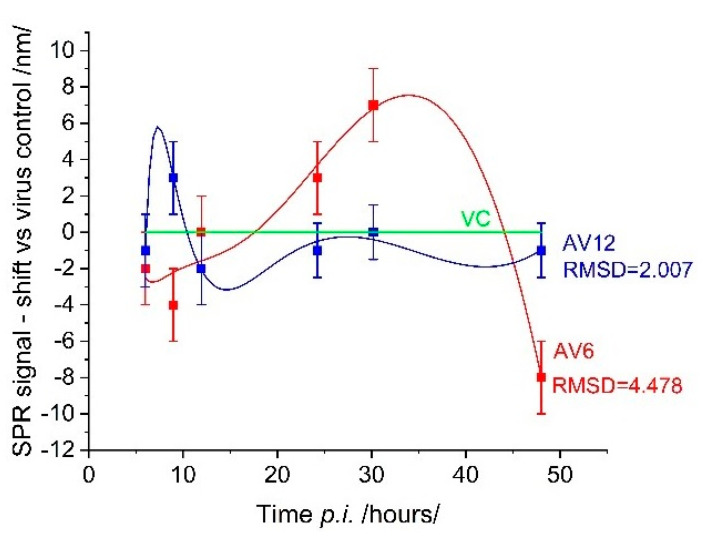
SPR evaluation of the inhibitory effect of AV6 and AV12 chalcons.

**Figure 8 sensors-25-03905-f008:**
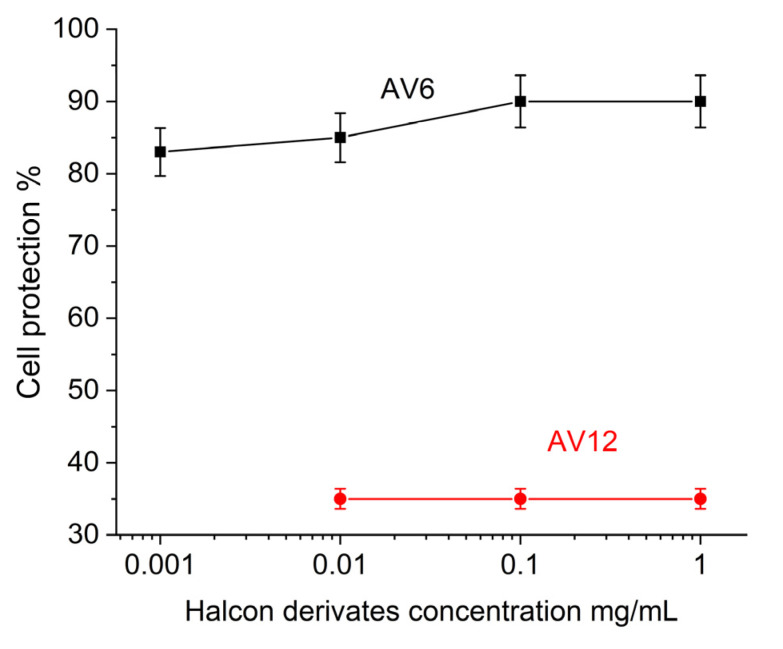
MTT evaluation of antiviral activity of AV6 and AV12 chalcon derivatives.

## Data Availability

Dataset available on request from the authors.

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
