# Peer review of "Does a SPR-Based Cell-Based Assay Provide Reliable Results on the Toxicity and Efficacy of Antiviral Drugs?"

_sensors, 2025, doi:10.3390/s25133905_

Round 1

Reviewer 1 Report

Comments and Suggestions for Authors

Petia Genova-Kalou et al. developed a grating-based surface plasmon resonance (SPR) technique and applied it to investigate the feasibility of cell-based antiviral drug screening experiments, confirming the efficacy and reliability of this method for cell analysis. However, the introduction of the article lacks sufficient detail.The paper was organized well and provided a solid data to support the point. The paper could be published in the revised version after carefully address the issues.

  1. In the introduction of the paper, it is stated that Surface Plasmon Resonance (SPR) technology is frequently employed to detect biomolecular interactions; however, recent advancements in SPR technology for biomolecule detection are not discussed.Hope for a more comprehensive analysis.
  2. In the introduction of the paper, it is stated that the cells are immobilized on the surface of the SPR chip, with a penetration depth ranging from 150 to 400 nm from the metal surface, which falls within the micrometer range. Please describe the specific impact of penetration depth on this work and its advantages.
  3. A comparison should be made among various recent cell analysis techniques, such as Ω-shaped fiber optic-based LSPR and Raman analysis technology, and the grating-based SPR cell analysis technology proposed in this paper. This comparison should clarify the advantages and disadvantages of each technique to illustrate the advancements represented by the technology used in this paper. The corresponding works should be cited.
  4. Most of the citations are from literature published over 10 years ago. It is hoped that more recent publications will be included to enhance the relevance and reliability of the citations.

Comments on the Quality of English Language

Petia Genova-Kalou et al. developed a grating-based surface plasmon resonance (SPR) technique and applied it to investigate the feasibility of cell-based antiviral drug screening experiments, confirming the efficacy and reliability of this method for cell analysis. However, the introduction of the article lacks sufficient detail.The paper was organized well and provided a solid data to support the point. The paper could be published in the revised version after carefully address the issues.

  1. In the introduction of the paper, it is stated that Surface Plasmon Resonance (SPR) technology is frequently employed to detect biomolecular interactions; however, recent advancements in SPR technology for biomolecule detection are not discussed.Hope for a more comprehensive analysis.
  2. In the introduction of the paper, it is stated that the cells are immobilized on the surface of the SPR chip, with a penetration depth ranging from 150 to 400 nm from the metal surface, which falls within the micrometer range. Please describe the specific impact of penetration depth on this work and its advantages.
  3. A comparison should be made among various recent cell analysis techniques, such as Ω-shaped fiber optic-based LSPR and Raman analysis technology, and the grating-based SPR cell analysis technology proposed in this paper. This comparison should clarify the advantages and disadvantages of each technique to illustrate the advancements represented by the technology used in this paper. The corresponding works should be cited.
  4. Most of the citations are from literature published over 10 years ago. It is hoped that more recent publications will be included to enhance the relevance and reliability of the citations.

Author Response

We would like to thank the reviewer for his positive review and for asking questions that helped us improve the quality of the paper.

  1. In the introduction of the paper, it is stated that Surface Plasmon Resonance (SPR) technology is frequently employed to detect biomolecular interactions; however, recent advancements in SPR technology for biomolecule detection are not discussed.Hope for a more comprehensive analysis.

In the new paragraph, (lines 48-57) we describe the factors that have influenced the development of SPR technology to date.

 First and foremost among these is the need to study the biochemistry of proteins and newly synthesised molecules included in new drugs. A variety of SPR techniques have been employed for the detection of different analytes under various conditions, with ligands immobilized using a range of methods. There are hundreds of papers and PhD theses devoted to these topics. It is challenging to provide a comprehensive overview of such a broad field in an article that does not focus on biomolecular interactions. Therefore, we direct the reader's attention to review articles.

2. In the introduction of the paper, it is stated that the cells are immobilized on the surface of the SPR chip, with a penetration depth ranging from 150 to 400 nm from the metal surface, which falls within the micrometer range. Please describe the specific impact of penetration depth on this work and its advantages.

The paragraph (lines 62-77) was modified giving more clear information.

The analysis provided in Figure 3 (lines 246-255) illustrates the benefits of enhanced penetration depth, showcasing higher levels of accuracy and sensitivity.

This result is not surprising. Theory and experiments have shown that increasing the SPR wavelength increases the penetration depth of the plasmon wave, which significantly improves sensitivity. This is why IR-SPR detection was developed: please refer to https://doi.org/10.1063/1.3116143

3. A comparison should be made among various recent cell analysis techniques, such as Ω-shaped fiber optic-based LSPR and Raman analysis technology, and the grating-based SPR cell analysis technology proposed in this paper. This comparison should clarify the advantages and disadvantages of each technique to illustrate the advancements represented by the technology used in this paper. The corresponding works should be cited.

In the new paragraph ( lines 73-85), we present a comparison of Raman spectroscopy of cells and SPR-based cell analysis.

At present, Ω-shaped fibre optic-based LSPR is used as a cytosensor. The detection of specific living cells is based on an immobilized ligand that interacts specifically with cell membrane receptors.

SPR-based cell assay is not intended to detect cells, but rather to track changes in their life status.

4. Most of the citations are from literature published over 10 years ago. It is hoped that more recent publications will be included to enhance the relevance and reliability of the citations.

As indicated in paragraph lines 48-57, the development of SPR technology has been driven by research into biomolecular interactions. Among SPR technologies, mentioned in the paragraph, only SPR imaging is widely used for cell analysis. Its application is systematised in:

Su, Yu-wen, and Wei Wang. "Surface plasmon resonance sensing: from purified biomolecules to intact cells." Analytical and bioanalytical chemistry 410 (2018): 3943-3951.

There have been no significant results published in this field recently.

It is only in recent years that cell analyses have become the subject of SPR detection due to the acceptance of the thesis that a physiologically relevant environment is necessary for evaluating the effectiveness of biochemical reactions. However, cell assay is still not a mainstream in SPR research. This explains the small number of articles in this field.

The latest results relating to an SPR-based cell assay are mentioned in the article (see ref. 14).

It is surprising that there is no review article on the SPR-based cell assay to date. Our team is in the process of presenting such an article in a few months.

The distinguished reviewer noted that the English language could be refined. A thorough review of the article has been conducted, resulting in the correction of several errors.

Reviewer 2 Report

Comments and Suggestions for Authors

The authors present an interesting study on cell assay-based drug cytotoxicity and grating-based SPR reliability for this type of measurement. The findings are interesting, and the article is well-written. However, I would suggest that the authors clarify a few questions, which would in turn benefit the article:

The measurement scheme would help in understanding how the measurement is performed. Currently, it is unclear how the sample is excited, and how the response is collected.
Was the spectrum of SPR measured? Why the angles of incidence were chosen from 41 degrees to 80? Are the measurements at large angles more convenient than at small angles (maybe there are some advantages)?
As the angle of incidence gives a different SPR spectrum, do you choose the excitation wavelength at the SPR intensity minima (the most shallow point)? How is the sensitivity of SPR affected by a slight wavelength deviation from the SPR dip minimum?

The article can be published after addressing these points.

Author Response

We would like to thank the reviewer for his positive review.

The measurement scheme would help in understanding how the measurement is performed. Currently, it is unclear how the sample is excited, and how the response is collected.

Lines 108-110 specify how SPR is excited:

It should be noted that, unlike all similar studies in the field, we use grating-based SPR. Since SPR is excited on the analyte side, it is difficult to use a controlled flow of the analyte.

Corrected: It is clarified that a grating is illuminated by light passing through the cells.

Was the spectrum of SPR measured?

Lines 134-136 specify that the spectrum of the reflected light is measured:

Spectral interrogation was employed to monitor SPR in the zero order of reflection using an Avasoft АVASPEC-ULS2048CL-EVO spectrometer.

Why the angles of incidence were chosen from 41 degrees to 80? Are the measurements at large angles more convenient than at small angles (maybe there are some advantages)?

Lines 125-130 reveal the reason for SPR excitation at two different angles:

The key advantage of a diffraction grating is that it allows straightforward tuning of the SPR wavelength by altering the incident angle, ranging from approximately 610 nm (at 80 deg) to around 1030 nm (at 41 deg). This allows the penetration depth of the plasmon wave field to be adjusted from about 300 nm to about 520 nm, and consequently the detection sensitivity to be varied.

As the angle of incidence gives a different SPR spectrum, do you choose the excitation wavelength at the SPR intensity minima (the most shallow point)? How is the sensitivity of SPR affected by a slight wavelength deviation from the SPR dip minimum?

As previously stated, a spectral interrogation method was employed to observe SPR in the light reflected from the grating. This method requires a polychromatic light source. We do not change the wavelength of the light source, only the angle of incidence. This changes the wavelength at which the SPR is excited, and consequently the detection sensitivity.

In this instance, sensitivity is determined by the sharpness of the SPR dip, which is, in turn, determined by the quality of the diffraction grating. The detection sensitivity is also determined by the sensitivity of the spectrometer. The factors that determine the sensitivity of surface plasmon resonance (SPR) detection are systematized in Shalabney, Atef, and Ibrahim Abdulhalim's "Sensitivity-Enhancement Methods for Surface Plasmon Sensors." Laser & Photonics Reviews 5, no. 4 (2011): 571–606.

Reviewer 3 Report

Comments and Suggestions for Authors

This study evaluates the use of grating-based Surface Plasmon Resonance (SPR) in cell-based assays for antiviral drug screening, comparing its performance with the traditional MTT assay. The authors developed a method to monitor SPR signals at discrete time intervals post-cell seeding, enabling the study of cytotoxicity and antiviral efficacy without continuous flow. Using Vero E6 cells and HCoV-229E virus, they tested chalcone derivatives (AV6 and AV12) and observed SPR signal changes in relation to cell coverage, cytotoxicity, and viral inhibition. The SPR results correlated strongly with MTT assay outcomes, but SPR provided earlier detection (before 48 h vs. 96 h in MTT). The study highlights the promise of grating-based SPR for high-sensitivity, label-free, and time-efficient drug screening.

  1. Figures 1–3: The seeding density is reported without units; please indicate the appropriate unit (e.g., cells/mL).
  2. Figure 4: The image quality is poor, and a higher magnification is needed to clearly observe the cell morphology and confluency.
  3. Figure 8: The MTT results lack statistical analysis, such as error bars and significance testing, which are essential for assessing data reliability.
  4. While SPR's early detection is a highlight, the manuscript does not quantify the limit of detection (LOD) or sensitivity range of the SPR setup in terms of cellular or viral response. A discussion or experiment addressing this would strengthen claims of high sensitivity.
  5. The paper suggests that the SPR signal reflects morphological and biochemical changes near the cell membrane but does not directly demonstrate this mechanistically. Could the authors provide additional supporting evidence (e.g., imaging, markers) or discuss limitations in differentiating cytopathic effects vs. antiviral activity?

Author Response

We would like to thank the reviewer for his positive review and for asking questions that helped us improve the quality of the paper.

1. Figures 1–3: The seeding density is reported without units; please indicate the appropriate unit (e.g., cells/mL).

Corrected

2. Figure 4: The image quality is poor, and a higher magnification is needed to clearly observe the cell morphology and confluency.

Corrected

3. Figure 8: The MTT results lack statistical analysis, such as error bars and significance testing, which are essential for assessing data reliability.

Corrected:

New paragraph (lines 339-345) provides information about MTT data processing.

4. While SPR's early detection is a highlight, the manuscript does not quantify the limit of detection (LOD) or sensitivity range of the SPR setup in terms of cellular or viral response. A discussion or experiment addressing this would strengthen claims of high sensitivity.

The new paragraph (lines 246-255) discusses the dynamics and time dependence of surface plasmon resonance (SPR) signals measured in identical slides in the visible and infrared (IR) regions. The analysis proves different detection sensitivities.

In SPR biorecognition applications, the limit of detection (LOD) is relatively easy to determine when solutions of different concentrations are used. This was demonstrated in the paper:

Dyankov, Georgi, Petia Genova-Kalou, Tinko Eftimov, Sanaz Shoar Ghaffari, Vihar Mankov, Hristo Kisov, Petar Veselinov, Evdokia Hikova, and Nikola Malinowski. "Binding of SARS-CoV-2 structural proteins to hemoglobin and myoglobin studied by SPR and DR LPG." Sensors 23, no. 6 (2023): 3346.

The LOD problem is complex in SPR-based cell assays. To our knowledge, no literature exists on LOD estimation..

We discuss these issues in the new paragraphs (lines 263-267, lines 374-383).

The figure below shows the detection sensitivity of the SPR slide used in the model experiments when treated with solutions of different refractive index (RI). Apparently, sensitivity in the NIR region is higher. This results in a lower LOD.

This experimental result has not been published. It should be viewed as personal information..

5. The paper suggests that the SPR signal reflects morphological and biochemical changes near the cell membrane but does not directly demonstrate this mechanistically. Could the authors provide additional supporting evidence (e.g., imaging, markers)

Morphological changes to the area of the cell membrane can be observed using AFM. Such studies were conducted in our previous work.

Genova-Kalou, Petia, Georgi Dyankov, Radoslav Marinov, Vihar Mankov, Evdokiya Belina, Hristo Kisov, Velichka Strijkova-Kenderova, and Todor Kantardjiev. "SPR-Based Kinetic Analysis of the Early Stages of Infection in Cells Infected with Human Coronavirus and Treated with Hydroxychloroquine." Biosensors 11, no. 8 (2021): 251.

The AFM image shown here depicts an area close to the membrane of an infected cell 24 hours post-infection (p.i.). Viruses (marked red) have just been expressed from the host cell and are still close to the cell membrane. The second replication cycle then begins with virus attachment to the cell membranes of new cells. Consequently, compaction of the cell membrane occurred and the refractive index increased. This coincided with an increase in the SPR signal observed after 30 hours.

The image demonstrates the morphology of infected cell - several virus particles (marked yellow) budding to the host cell membrane.

The present study did not include similar AFM observations, but given that the object of study is identical, no other result can be expected.

…. or discuss limitations in differentiating cytopathic effects vs. antiviral activity?

Antiviral studies are always conducted at non-toxic concentrations of the compound, as stated  in lines…  Therefore, cytotoxic effects are not expected.

Round 2

Reviewer 1 Report

Comments and Suggestions for Authors

All issues have been addressed. The version of the manuscript was suggested to be accepted.